# Factors Influencing the Duration of Termination of Pregnancy for Fetal Anomaly with Mifepristone in Combination with Misoprostol

**DOI:** 10.3390/jcm12030869

**Published:** 2023-01-21

**Authors:** Theresa Reischer, Iris Limbach, Anja Catic, Katrin Niedermaier, Veronica Falcone, Gülen Yerlikaya-Schatten

**Affiliations:** Division of Obstetrics and Feto-Maternal Medicine, Department of Obstetrics and Gynaecology, Medical University of Vienna, 1090 Vienna, Austria

**Keywords:** termination of pregnancy, feticide, fetal malformation, genetic abnormalities, induction of labor, mifepristone, misoprostol, fetal expulsion

## Abstract

This study’s aim was to determine relevant factors that influence the time interval between first induction and fetal expulsion in late termination of pregnancy (TOP) and TOP after previous feticide for severe fetal malformation with a mifepristone–misoprostol regime. This retrospective study included 913 TOPs from a single tertiary care referral center. In 197 out of 913 TOPs, a previous feticide had been performed due to advanced gestational age (after 22 + 0 weeks of gestation). Induction was accomplished using 600 mg mifepristone followed by 400 μg misoprostol. The interval between first induction with misoprostol and fetal expulsion was examined. Univariate and multivariate logistic regression analysis were used to predict an induction interval of 12 h or less. The median gestational age at induction of labor was 18.9 weeks of pregnancy. In 487 (53.3%) cases women delivered within 12 h; in 344 (37.7%) cases the induction interval was between 12 h and 36 h. In 82 (9%) cases induction took longer than 36 h. Factors that were significantly associated with a delivery duration of <12 h were a lower gestational age at induction (OR 0.87; 95% CI 0.84–0.89; *p* < 0.001) and a history of at least one previous vaginal delivery (OR 1.57; 95% CI 1.20–2.05; *p* < 0.001). Factors that had no impact included previous cesarean section, performing feticide before induction and maternal age. Maternal BMI showed a non-significant trend.

## 1. Introduction

The implementation of screening programs, improved ultrasound technology and new genetic testing methods have led to an improved detection rate for numerous fetal malformations during pregnancy [1,2,3]. Consequently, the issue of late termination of pregnancy and possible feticide has increased in importance, although the majority of terminations, up to 90%, are still performed in the first trimester, less than 10% in the second trimester and the rest in the third trimester [4].

Different regimens for second trimester termination of pregnancy have been published. Most of these are based on misoprostol or gemeprost used alone, or mifepristone combined with misoprostol. The latter seems to be the most effective method regarding the highest efficacy and shortest abortion time interval [5]. However, the precise timing from the start of initiation to complete expulsion is difficult to determine. Furthermore, there are only occasional data regarding various factors that may influence the time interval, such as gestational age at induction of labor (IOL), parity, previous cesarean section and BMI [6,7,8,9,10]. In previous studies by Dickenson et al. and Vitner et al., gestational age and parity were identified as additional factors that have an influence on the induction-to-abortion time. Previous vaginal deliveries and early gestational age shortened the time of expulsion [8,11]. However, the case numbers were low in the work by Vitner et al., and both studies reported just the use of misoprostol alone [8,11].

The aim of this study was to determine relevant maternal characteristics and factors that influence the time interval between first induction and fetal expulsion in early and late termination of pregnancy (TOP) and TOP after previous feticide for severe fetal malformation and/or chromosomal abnormality with a mifepristone combined with misoprostol regime. 

## 2. Materials and Methods

### 2.1. Study Population

This was a retrospective study of prospectively collected data regarding 913 singleton pregnancies ending in termination of pregnancies (TOP) at the Department of Obstetrics and Gynecology at the Medical University of Vienna, a single tertiary care referral center, between 2007 and 2020. The inclusion criteria were pregnant women with prenatally diagnosed fetal malformations and/or genetic defects who opted for termination of pregnancy. Pregnancies with selective feticide and those ending in a miscarriage were excluded. After 22 + 0 weeks of gestation, feticide was offered, whereas TOP after 23 + 0 weeks was only possible with previous feticide to avoid live births, except for rare cases with lethal conditions. Feticide was performed via intracardiac injection of either potassium chloride or lidocaine; in some cases, due to fetal position, the injection was performed through the umbilical cord. The pregnancies were terminated for severe fetal anomalies and/or genetic abnormalities. Every TOP for fetal abnormalities was discussed by an internal clinical committee. Labor was induced with mifepristone and misoprostol [12,13]. All women with and without previous cesarean section were treated with 600 mg mifepristone. Patients with a previous feticide received 600 mg mifepristone within 30 min after the procedure; 24 h after taking mifepristone, 400 μg misoprostol was started orally. Administration of misoprostol was repeated every 3 h until regular contractions were recorded. For those patients who had a previous cesarean section, misoprostol was administered slightly differently on the first day (50 μg, 100 μg and 200 μg) and increased the next day to 400 μg every 3 h until regular contractions began.

General medical history and maternal characteristics were extracted from our perinatal database retrospectively.

### 2.2. Patient Characteristics

Patient characteristics that were evaluated included maternal age (years) and body-mass-index (BMI; kg/m^2^). Obstetric history included parity (n), history of vaginal delivery (yes/no), history of cesarean section (yes/no), gestational age at the time of induction of labor, and the time span between the start of induction and fetal expulsion.

### 2.3. Statistical Analysis

For statistical analysis, metric variables were presented as mean +/- SD in cases of standard distribution or median and IQR. Metric variables were compared using the Student’s *t*-test or the Mann–Whitney U-test when variables did not follow standard distribution. A two-sided *p*-value of <0.05 was set for statistical significance. Univariate logistic regression was used to assess factors associated with fetal expulsion within 12 h. Subsequently, a multivariate logistic regression model was stepwise fitted to test variables showing a trend with a *p*-value < 0.1 after univariate analysis. Covariates included gestational age at induction, previous vaginal delivery, Body-Mass-Index, dosage (after cesarean section) and previous feticide. Statistical analyses were performed using IBM SPSS Statistics Version 23.

## 3. Results

The average maternal age of women undergoing TOP was 32.1 years (SD ± 6.2), ranging from 13 to 49 years.. Overall, 716 fetuses had a TOP without feticide at an average of 16.8 weeks of gestation. In 197 cases, feticide was performed; reductions and selective feticides in multiple pregnancies were excluded. Furthermore, 6 out of 919 women were excluded due to cesarean section. One cesarean was performed due to failed IOL, and two due to heavy bleeding during induction time. Two cesarean sections were planned and carried out because of known placenta previa, and another due to three previous cesarean sections. Therefore, with three unplanned cesarean sections after IOL, we had a success rate of 99.7% with the mifepristone combined with misoprostol regime. Maternal characteristics for the 913 pregnancies evaluated and used for further calculations are shown in Table 1.

The gestational age was less than 14 + 0 weeks of pregnancy (WoP) in 134 (14.7%) cases. In 362 (39.6%) cases, the gestational age was between 14 + 0 and 19 + 6 WoP, and in 292 (32.0%) cases, it was between 20 + 0 and 23 + 6 WoP. Late termination of pregnancy after 24 + 0 WoP accounted for 125 (13.7%) cases. Median gestational age at induction of labor was 18.9 WoP, ranging from 10 to 37 WoP. In 487 (53.3%) cases women delivered within 12 h after intake of misoprostol. In 344 (37.7%) cases the induction interval was between 12 h and 36 h, and in 82 (9%) cases induction took longer than 36 h (Figure 1).

To evaluate possible factors influencing the interval of induction for further evaluation, cases were grouped in two categories, delivery within 12 h and expulsion after more than 12 h. Possible factors influencing the interval of induction that were used for analyses included gestational age at induction, previous vaginal delivery, previous cesarean section, feticide before IOL and maternal BMI and age.

First, we analysed the influence of gestational age at induction. We showed that almost 77% of cases in the first trimester delivered within 12 h after IOL with mifepristone combined with misoprostol, and that just 23.1% of cases delivered after 12 h. In contrast, approximately 28% of cases delivered within 12 h after IOL in the third trimester. In the second trimester, half of the cases delivered within 12 h after IOL. The results are summarized in Table 2. 

To assess a possible influence of feticide on the time interval from induction to expulsion, we compared only cases of TOP after 22 WoP, which revealed no significant difference (OR 0.99; 95% CI 0.55–1.76; *p* > 0.05).

The results of the univariate analyses revealed that in addition to gestational age at induction, history of at least one previous vaginal delivery and dosage of misoprostol were statistically significant factors associated with the interval of induction. Cases with induction at an earlier stage in pregnancy and cases with previous vaginal delivery were more likely to experience expulsion within 12 h. The univariate analysis revealed that a lower dosage of misoprostol due to previous cesarean section was associated with delivery after 12 h of induction. Maternal BMI showed a non-significant trend, whereas maternal age did not affect the interval of induction. Multivariate statistical logistic regression was used to predict expulsion within 12 h after induction; only gestational age at induction and history of previous vaginal delivery were significant contributors (covariates included BMI and misoprostol dosage) (Table 3).

## 4. Discussion

In this retrospective study, univariate and multivariate logistic regression analyses were used to evaluate possible factors influencing the interval of induction of labor (IOL) for fetal malformation and genetic abnormalities. Factors that had a significant impact on the duration of delivery (induction–expulsion interval of 12 h or less) were gestational age at induction and history of vaginal delivery. Considering these two factors, when informing patients about induction procedure it is possible to anticipate fetal expulsion within less than 12 h after IOL.

These results are comparable to previous works on this topic [14,15,16]. Ashok et al. assessed the IOL via administration of 600 mg mifepristone and 400 mg misoprostol with 97.1% of cases experiencing fetal expulsion within 15 h [14]. Another study showed that the average interval from induction to expulsion was 12.5 +/−7.5 h in TOPs in the second and third trimesters [16]. It is noteworthy that this interval was remarkably shorter in multiparas and cases of fetal death. In our analysis, intrauterine fetal death (IUFD) was excluded, and only cases that had an active termination of pregnancy with or without feticide for fetal malformation were included. There are no meaningful data published yet regarding feticide and duration of induction. Nevertheless, previous feticide did not affect the time interval between first induction and fetal expulsion in TOP (after adjusting for gestational age at IOL), and was not a relevant covariate in this investigation. Overall, it seems that there were correlations between the mifepristone and misoprostol regime and gestational age at IOL and duration of fetal expulsion. However, a previous study by Lo et al. reported contrary results [10]. No relationships between using misoprostol and gestational age or duration of expulsion have been observed; however, pregnancy termination before 17 weeks was associated with a higher chance of incomplete abortion [10].

Another interesting covariate is the BMI of the pregnant women. In our study population, a non-significant trend regarding maternal weight was observed. This means, the higher the maternal weight, the greater the likelihood that induction would take longer than 12 h. Helmig et al. conducted a study to investigate whether the cumulative dose of misoprostol required for IOL is associated with BMI. Their results showed that time to delivery and the risk of cesarean section increased with rising BMI class [17]. In addition, Freret et al. identified BMI as a risk factor for unsuccessful induction, especially BMIs of 40 or higher [18]. However, both studies refer to pregnancy inductions at term [17,18]. Obesity, among other things, has been associated with increased risk of poor reproductive outcomes, including poor pregnancy outcomes.

Cesarean section did not seem to influence the interval of induction to complete fetal expulsion. Furthermore, previous cesarean sections did not have a negative impact on the length of delivery, despite lower dosages of misoprostol. A previous work also described a 95.72% success rate of misoprostol administration for pregnancy termination in a retrospective study including 678 subjects [19]. We can report similar results. In our study population, 3 out of 916 women had a cesarean section after IOL, which is a success rate of 99.7%. However, these cases were excluded from further analyses. Another study analyzed the efficacy and safety of misoprostol for second trimester termination of pregnancy among women with one or more previous cesarean sections [7]. A median time interval from IOL to fetal expulsion of 9 h was observed [7]. However, regime and use of misoprostol after cesarean section varied widely, making comparisons difficult.

Overall, the use of mifepristone and misoprostol for pregnancy termination appeared to be effective in all three trimesters. Factors influencing the duration of termination of pregnancy for fetal anomaly with mifepristone in combination with misoprostol were gestational age at induction and previous vaginal delivery. A previous study showed that IOL with a mifepristone–misoprostol regime shortened the induction to delivery time; however, the effect depended on when mifepristone was administered [20]. If mifepristone was administered 1 day prior to induction, delivery occurred within 24 h after the first induction in more than 90% of cases. If mifepristone was administered later, the effect was less pronounced [20]. This working group evaluated termination in second and third trimesters with a mifepristone–misoprostol regimen, but there was no information regarding the number of possible feticides performed and their possible impact on induction–expulsion duration [20].

The diagnosis of fetal anomaly is devastating for future parents. From the time of diagnosis until induction of labor, these women feel alone and vulnerable, and many suffer long-term emotional and psychological sequela. Reports of pain perception during the termination of pregnancy in this particular group of patients are meager. It is believed that parenteral opioids may be an option, as the negative effect on the fetus can be neglected [21]. A study by Mazouni et al. could not show any differences in outcomes in women who received epidural analgesia and those who received parenteral opioids [22]. However, another study observed that labor with epidural analgesia had a shorter first stage, faster rate of cervical dilation, and unchanged second stage duration compared with women receiving parenteral opioids [23]. In 2015, a prospective, double-blind, randomized trial was performed in which women were randomly assigned to receive continuous epidural infusion or programmed intermittent epidural bolus for pain relief during termination of pregnancy. Both techniques showed low pain scores that were comparable between groups [24]. Most centers with (later) termination offered maternal anxiolytic and/or sedative medications for feticide procedures and deliveries, whereas epidural techniques were the most common [25]. Nevertheless, there are no standardized recommendations regarding anesthesia and analgesia for late terminations.

The limitation of this study was its retrospective setting, with data collected via record review and no prospective information. However, the strengths of the study include the large number of cases and the evaluation of late pregnancy termination with a previous feticide, i.e., TOP after 22 weeks of pregnancy, whereas the majority of the literature reported only up to 23/24 weeks of gestation.

## 5. Conclusions

In conclusion, a multivariate logistic regression analysis identified gestational age and previous history of vaginal delivery as relevant contributors regarding the time between IOL and fetal expulsion. Feticide before induction of labor, BMI and previous cesarean section were not significant covariates.

## Figures and Tables

**Figure 1 jcm-12-00869-f001:**
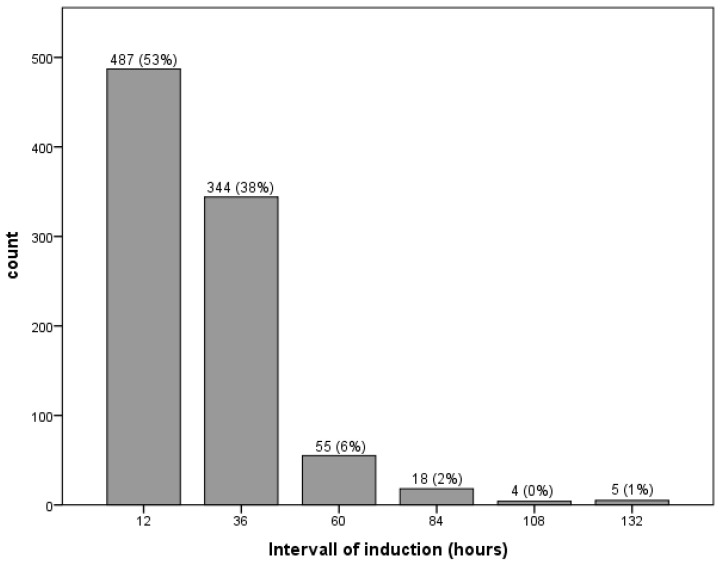
Interval between first induction and fetal expulsion.

**Table 1 jcm-12-00869-t001:** Maternal characteristics.

	N (%)	Min	Max	Mean ± SD	Median
Maternal age (years)		13	49	32.1 ± 6.2	
BMI		16	45	24.3 ± 4.7	
Gravida		1	14		2
Para		0	8		1
Previous vaginal delivery					
Yes	536 (58.7%)				
No	377 (41.3%)				
Previous C-Section					
Yes	139 (15.2%)				
No	774 (84.8%)				
Conception					
Spontaneous	845 (93%)				
ART	68 (7%)				

**Table 2 jcm-12-00869-t002:** Interval between induction of labor and fetal expulsion depending on the pregnancy trimester.

	Expulsion within 12 h n (%)	Expulsion > 12 h n (%)
First Trimester	103 (76.9%)	31 (23.1%)
Second Trimester	372 (50.5%)	364 (49.5%)
Third Trimester	12 (27.9%)	31 (72.1%)

**Table 3 jcm-12-00869-t003:** Univariate and multivariate logistic regression analysis predicting an induction–expulsion interval of 12 h or less.

**Univariate Analysis**	**OR**	**95% CI**	***p*-Value**
Gestational age at induction (weeks)	0.87	0.84–0.89	<0.001
Previous vaginal delivery	1.57	1.20–2.05	<0.001
Body-Mass-Index (kg/m^2^)	0.98	0.95–1.00	0.078
Misoprostol dosage *	0.45	0.31–0.65	<0.001
Feticide	0.35	0.25–0.49	<0.001
**Multivariate Analysis**	**OR**	**95% CI**	***p*-Value**
Gestational age at induction	0.83	0.80–0.86	<0.001
Previous vaginal delivery	2.65	0.45–1.15	<0.001

* In cases with a previous cesarean section, a lower dosage was used during the first day of induction.

## Data Availability

Data presented in this study are available on request from the corresponding author. Data are not publicly available due to data privacy requirements.

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
