# Peer review of "Factors Influencing the Duration of Termination of Pregnancy for Fetal Anomaly with Mifepristone in Combination with Misoprostol"

_jcm, 2023, doi:10.3390/jcm12030869_

Round 1

Reviewer 1 Report

General comments: This manuscript describes the clinical experience of a single tertiary referral center in providing later terminations using mife/miso regimens. The outcome of interest was fetal expulsion within 12h of initiation of induction. The manuscript provides extremely minimal interpretation of the results for the reader and thus requires revision. Additionally, there are many places where methods should be clarified. With appropriate revisions, this work will provide important information related to the timing of termination using mife/miso regimens, especially in later terminations. Specific comments are below.

Abstract

1.       Line 15: Can the weeks of gestation be described to qualify “advanced gestational age”?

2.       Line 16: Can “first induction” be defined?

3.       Lines 20-24: Please provide further details for readers. In what direction were these results? How large were the effect sizes?

4.       Lines 23-24: This repeats the information provided in lines 20-21.

Introduction

5.       Lines 43-45: Can the directions of these associations be described?

Materials and methods

6.       Line 56: If all 913 TOPs were of singletons, and thus singleton pregnancy was an inclusion criteria, this could be mentioned in the first line (line 54).

7.       Lines 57-58: Was it an inclusion criteria that all terminations be for severe anomalies/abnormalities?

8.       Line 63: What does “Induction/Procedure” refer to?

9.       Line 64: How were “regular contractions” defined?

10.   Lines 70-72: The inclusion criteria should be moved to the beginning of the study population paragraph, and duplicative information below should be deleted.

11.   Line 83: What are “metric” variables? Maybe “continuous”?

12.   Line 86: What factors were assessed as covariates? How was this list decide upon?

13.   Line 86: This is not an actual prediction model, correct? Maybe “…to assess factors associated with fetal expulsion…”?

Results

14.   Line 94: These 716 fetuses should be described as those without feticide, correct?

15.   Line 95: This information has been provided in the Methods and does not need to be repeated here.

16.   Lines 96-99: This sounds like six cesarean sections? This should be revised for clarity. Also, given that cesarean section is not the main outcome of interest in the study, it may make sense to move this section below the section about time to fetal expulsion.

17.   Line 105: The last paragraph of the Introduction states that the study purpose is to look at induction in late termination of pregnancy, but first trimester terminations are being included. Either the stated purpose or the study population should be updated to make these consistent.

18.   Table 1: Median gravidity and parity is not very informative. Readers may prefer to see the proportion of patients who are nulligravid/parous vs. multigravid/parous.

19.   Figure 1: Can proportions be added above the bars?

20.   Lines 129-130: Are these results shown anywhere? Even if the result is not statistically significant, it would be nice for the reader to see the results for themselves.

21.   Lines 134-140: It would be helpful to actually interpret these results for the readers. In which direction were these factors associated? What was the effect size?

22.   Line 136: “Influencing” implies causality, whereas these are just associations.

23.   Line 137: Why is maternal age not included in Table 3?

24.   Lines 137-138: What is the multivariable model controlling for?

25.   Lines 139-140: Is it actually the case that obstetrical history was separated into prior vaginal delivery and no prior cesarean delivery? It sounds elsewhere that this is not the case. It appears that “spontaneous delivery” and “vaginal delivery” are sometimes used interchangeably when they are not the same thing.

26.   Table 3: What are the units for gestational age at induction, BMI, and dosage (of what?)? These should be included, as otherwise it is impossible for the reader to interpret these results. Even if they are not significant, all of the variables should be shown in the multivariable analysis. Is the model specified correctly for the outcome (i.e., is expulsion prior to 12 hours chosen as the outcome as opposed to expulsion after 12 hours)? These results show that increasing gestational age is associated with increased odds of expulsion prior to 12 hours, which seems counter to the data presented in Table 2.

Discussion

27.   Lines 149-150: Again, some interpretation of the results is needed here.

28.   Lines 151-152: By combining it how? Please provide interpretation for the reader.

29.   Lines 186-187: Were the excluded or were they just not eligible for the outcome of vaginal delivery?

Author Response

Reviewer 1:

Thank you very much for reviewing our manuscript and the positive evaluation of our work. We appreciate the encouragements and constructive comments as well as the efforts regarding the improvement of the manuscript.

In order to facilitate the review process, we provided a point-by-point response to each of the comments and marked all changes in the revised manuscript.

Abstract

Line 15: Can the weeks of gestation be described to qualify “advanced gestational age”?

Response:  After 22+0 weeks of gestation feticide was offered, whereas TOP after 23+0 weeks was only possible with previous feticide to avoid live births.

 Line 16: Can “first induction” be defined?

Response: We counted as “first induction” the administration of 400mg misoprostol. We have added the information to the Abstract.

Lines 20-24: Please provide further details for readers. In what direction were these results? How large were the effect sizes?

Response: We added that information briefly as requested

 Lines 23-24: This repeats the information provided in lines 20-21.

Response: Thank you for your remark. We removed the repetition.

Introduction

Lines 43-45: Can the directions of these associations be described?

Response: This was added, as suggested

Materials and methods

Line 56: If all 913 TOPs were of singletons, and thus singleton pregnancy was an inclusion criteria, this could be mentioned in the first line (line 54).

Response: We have included the information now in the first line

Lines 57-58: Was it an inclusion criterion that all terminations be for severe anomalies/abnormalities?

 Response: All terminations of pregnancies performed at the department of gynecology and obstetrics at the medical university of Vienna were medically indicated terminations of pregnancies. According to the Austrian law a medical termination of pregnancy is just possible in case of a severe fetal malformation.

Line 63: What does “Induction/Procedure” refer to?

Response: We apologize for the inaccurate description. It refers to the administration of misoprostol; this was clarified now in the text

Line 64: How were “regular contractions” defined?

Response: Contractions are considered regular when the duration and frequency are stable over a period of time. 

Lines 70-72: The inclusion criteria should be moved to the beginning of the study population paragraph, and duplicative information below should be deleted.

 Response: Thank you for the good advice, this has been adapted

Line 83: What are “metric” variables? Maybe “continuous”?

Response: Metric variables are either measured or counted. Examples would be the weight and age of subjects.

Line 86: What factors were assessed as covariates? How was this list decide upon?

Response: Variables used for the multiple regression models were specified and the method of selection was described

Line 86: This is not an actual prediction model, correct? Maybe “…to assess factors associated with fetal expulsion…”?

Response: Thank you for your suggestion; this was adapted as suggested

Results

Line 94: These 716 fetuses should be described as those without feticide, correct?

 Response: Thank you for your comment, this has been corrected

Line 95: This information has been provided in the Methods and does not need to be repeated here.

Response: The sentence was removed as you suggested.

Lines 96-99: This sounds like six cesarean sections? This should be revised for clarity. Also, given that cesarean section is not the main outcome of interest in the study, it may make sense to move this section below the section about time to fetal expulsion.

 Response: Thank you for this important remark. The numbers of c-sections were corrected as suggested. As all the c-section were also excluded for further calculation, we believed it would be important to explain it in the beginning.

Line 105: The last paragraph of the Introduction states that the study purpose is to look at induction in late termination of pregnancy, but first trimester terminations are being included. Either the stated purpose or the study population should be updated to make these consistent.

Response: The aim of the study was corrected as follows: “The aim of this study was to determine relevant maternal characteristics and factors that influence the time interval between first induction and fetal expulsion in early and late termination of pregnancy (TOP) and TOP after previous feticide for severe fetal malformation and/or chromosomal abnormality with mifepristone combined with misoprostol regime.

Table 1: Median gravidity and parity is not very informative. Readers may prefer to see the proportion of patients who are nulligravid/parous vs. multigravida/parous.

Response: Thank you for this important comment. The information is already contained in the table: 41.3% nullipara whereas 58.7% already had a previous vaginal delivery.

Figure 1: Can proportions be added above the bars?

Response: Thank you for your comment. We labelled the bars with absolute numbers as suggested by reviewer 2 and percentage was added too.

Lines 129-130: Are these results shown anywhere? Even if the result is not statistically significant, it would be nice for the reader to see the results for themselves.

 Response: The results of the calculation were added to text.

Lines 134-140: It would be helpful to actually interpret these results for the readers. In which direction were these factors associated? What was the effect size?

 Response: Thank you for this important question. Information was added to the text and details are shown in table 3. The reference to the table was missing before and also has been added.

Line 136: “Influencing” implies causality, whereas these are just associations.

 Response: The term influencing was replaced by the term associated with.

Line 137: Why is maternal age not included in Table 3?

 Response: Thank you for the question. Table 3 only includes variables with a significant association or at least a trend with a p value of < 0.1. There was no trend or significant association between maternal age and interval of induction as described in line 154.

Lines 137-138: What is the multivariable model controlling for?

 Response: The information was added to the text for clarification.

Lines 139-140: Is it actually the case that obstetrical history was separated into prior vaginal delivery and no prior cesarean delivery? It sounds elsewhere that this is not the case. It appears that “spontaneous delivery” and “vaginal delivery” are sometimes used interchangeably when they are not the same thing.

Response: For obstetrical history previous vaginal delivery was recorded. As the history of previous cesarean determined the dosage of misoprostol, this information was recorded as well, this was described in line 75-77. The terms considering the mode of delivery were checked and corrected throughout the manuscript.

 Table 3: What are the units for gestational age at induction, BMI, and dosage (of what?)? These should be included, as otherwise it is impossible for the reader to interpret these results. Even if they are not significant, all of the variables should be shown in the multivariable analysis. Is the model specified correctly for the outcome (i.e., is expulsion prior to 12 hours chosen as the outcome as opposed to expulsion after 12 hours)? These results show that increasing gestational age is associated with increased odds of expulsion prior to 12 hours, which seems counter to the data presented in Table 2.

Response: Thank you for this important remark. All units of variables were added to the caption of the table. We corrected the calculations and the outcome now is expulsion within 12h. All calculations have been repeated and corrected. The multivariate model included all covariates and we chose to demonstrate the relevant ones – information about the variables used in the multivariate analysis was added to the text.

Discussion

Lines 149-150: Again, some interpretation of the results is needed here.

Lines 151-152: By combining it how? Please provide interpretation for the reader.

Response: we have changed the introduction of the discussion section as follows to make it easier for the reader to understand:

“ In this retrospective study univariate and multivariate logistic regression analysis were used to evaluate possible factors influencing the interval of induction of labor (IOL) for fetal malformation and genetic abnormalities. Factors that had a significant impact on the duration of delivery (induction-expulsion interval of 12h or less) were gestational age at induction and history of vaginal delivery without previous caesarean section. Taking these two factors into account, i.e. gestational age at induction and history of vaginal delivery, when informing patients about induction procedure”

Lines 186-187: Were the excluded or were they just not eligible for the outcome of vaginal delivery?

Response: Thank you for this important question. These cases were excluded and the information was added for clarification in line 113.

Reviewer 2 Report

I read with great interest the manuscript. In my opinion, the topic is interesting enough. Nevertheless, authors should clarify some points and improve the discussion, as suggested below.

- line 56-57: this is a result, not a method.

-line 60: explain feticide. How was it performed? 

-figure 1 is not clear. Please change it in a more readable one (e.g. the absolute values).

- Discussion can be improved citing: the pain related to the procedure; the importance to reduce the procedural time for psychological impact.

Author Response

Thank you very much for reviewing our manuscript and the positive evaluation of our work. We appreciate the encouragements and constructive comments as well as the efforts regarding the improvement of the manuscript.

In order to facilitate the review process, we provided a point-by-point response to each of the comments and marked all changes in the revised manuscript.

line 56-57: this is a result, not a method.

Response: The sentence has been moved to the results section as suggested

line 60: explain feticide. How was it performed? 

Response:  Feticide was performed with intracardiac injection of either potassium chloride or lidocaine, in some cases due to fetal position, the injection was performed through the umbilical cord.

This information was added to the methods section.

figure 1 is not clear. Please change it in a more readable one (e.g. the absolute values).

Response: Thank you for this helpful suggestion; we labelled the bars with absolute values and percentage and hope it is more readable now

Discussion can be improved citing: the pain related to the procedure; the importance to reduce the procedural time for psychological impact.

Response: Thank you for this important comment. We added this topic to the discussion.

The diagnosis of fetal anomaly is a devastating one for the future parents. From the timepoint of diagnosis until induction of labor those women feel alone and vulnerable and many suffer long-term emotional and psychological sequela. Reports of pain perception during the termination of pregnancy in this particular group of patients are very meager. It is believed that parenteral opioids may be an option since the negative effect on the fetus can be neglected [21]. A study by Mazouni et al. could not show any differences in outcomes in women who received epidural analgesia and those who received parenteral opioids [22], whereas another study observed labor with epidural analgesia shorter first stage, faster rate of cervical dilation, and unchanged second stage duration compared with women receiving parenteral opioids [23]. In 2015 a prospective, double-blind, randomized trial was performed where women were randomly assigned to receive continuous epidural infusion or programmed intermittent epidural bolus for pain relief during termination of pregnancy. Both techniques showed low pain scores which were comparable between groups [24]. However, most centers with (later) termination offer maternal anxiolytic and or sedative medications for feticide procedure and for delivery, whereas epidural techniques were the most common [25]. Nevertheless, there are no standardized recommendations regarding anesthesia and analgesia for these late terminations.

Reviewer 3 Report

The aim of the study entitled: “ Factors influencing the duration of termination of pregnancy for fetal anomaly with mifepristone in combination with misoprostol” was to determine the relevant maternal characteristics and factors that influence the time interval between first induction and fetal expulsion in late termination of pregnancy (TOP) and TOP after the previous feticide for severe fetal malformation and/or chromosomal abnormality with mifepristone combined with misoprostol regime.

Although the manuscript's findings are valuable and interesting, some obstacles prevent the manuscript from being published in its current form.

These, among others, include:

Materials and Methods

In lines 54-55… This was a retrospective study of prospectively collected data of 913 terminations of pregnancies (TOP) at a single tertiary care referral center between 2007 and 2020.- Please specify the name of the tertiary care referral center at which the study is conducted.

In lines 63-64… Induction/Procedure was repeated every 3 h until regular contractions were recorded.- This probably means that misoprostol was administered repeatedly every 3 h  until regular contractions were recorded. If so, please modify it.

Line 85… Metric variables were compared using students t-test Mann-Whitney U-test in case of variables not following standard distribution.- Please correct the sentence ( and should be added after the t-test). 

Also, the variables used in the multivariate logistic regression should be described.

Results

Line 96… In our study population 5 out of 916 women had a cesarean section.- The numbers (i.e., 916? Should it be 913) should be checked or explained.

The numbers for the Previous spontaneous delivery in table 1 (533 yes; 374 no---total 907) should be checked or explained.

Table 2 should be mentioned in the manuscript text. The same applies to table 3.

Please use the term TOP or ToP (line 130) for the termination of pregnancies, not both.

Is it possible to describe the timeframe (pregnancy trimester) for 197 cases of performed feticides?

Discussion

Lines 195-197  A previous study showed that  IOL with mifepristone – misoprostol regime shortens the induction to delivery time but the effect depends on when mifepristone has been given.- The reference is missing.

A minor revision of the manuscript is suggested.

Author Response

Thank you very much for reviewing our manuscript and the positive evaluation of our work. We appreciate the encouragements and constructive comments as well as the efforts regarding the improvement of the manuscript.

In order to facilitate the review process, we provided a point-by-point response to each of the comments and marked all changes in the revised manuscript.

Materials and Methods

In lines 54-55… This was a retrospective study of prospectively collected data of 913 terminations of pregnancies (TOP) at a single tertiary care referral center between 2007 and 2020.- Please specify the name of the tertiary care referral center at which the study is conducted.

Response: At the department of obstetrics and gynecology at the medical university of Vienna, a single tertiary care referral center between 2007 and 2020

 The information was added as suggested.

In lines 63-64… Induction/Procedure was repeated every 3 h until regular contractions were recorded.- This probably means that misoprostol was administered repeatedly every 3 h  until regular contractions were recorded. If so, please modify it.

 Response: Thank you for this important suggestion this was corrected

Line 85… Metric variables were compared using students t-test Mann-Whitney U-test in case of variables not following standard distribution. - Please correct the sentence (and should be added after the t-test). 

Also, the variables used in the multivariate logistic regression should be described.

Response: Thank you for your helpful comment; We corrected the sentence as suggested

Results

Line 96… In our study population 5 out of 916 women had a cesarean section. - The numbers (i.e., 916? Should it be 913) should be checked or explained.

Response: We apologize for the wrong number given in the text. The typo was corrected.

The numbers for the Previous spontaneous delivery in table 1 (533 yes; 374 no---total 907) should be checked or explained.

Response: Thank you for the important comment, this has been corrected.

Table 2 should be mentioned in the manuscript text. The same applies to table 3.

Response: Thank you for this important remark – table 2 and table 3 are mentioned in the text now

Please use the term TOP or ToP (line 130) for the termination of pregnancies, not both.

Response: This has been corrected.

Is it possible to describe the timeframe (pregnancy trimester) for 197 cases of performed feticides?

Response: This information was added in the methods section: “After 22+0 weeks of gestation feticide was offered, whereas TOP after 23+0 weeks was only possible with previous feticide to avoid live births….”

Discussion

Lines 195-197 A previous study showed that IOL with mifepristone – misoprostol regime shortens the induction to delivery time but the effect depends on when mifepristone has been given. - The reference is missing.

 Response: We apologize for the missing reference. It has been added now.

Round 2

Reviewer 1 Report

General comments: The authors have revised their manuscript to provide greater clarity in their methods and interpretation of their results. I have a few remaining minor comments.

Abstract

1.       Lines 21-23: Thank you for clarifying the direction of the results. Because “positive” results may be read as increasing the outcome (delivery duration), it may be better to say something like “factors that were significantly associated with delivery duration of <12 hours were…”  

Introduction

2.       Lines 47: A minor comment, but I believe “whereby” is the intended word as opposed to “whereas.”

Results

3.       Table 1: Is the 41.3% just for vaginal delivery or for any delivery? If vaginal, it won’t be possible for the reader to determine how much overlap exists between the vaginal and cesarean groups in order to determine how many patients were nulliparous. If the authors are concerned only with prior vaginal delivery, they should avoid the term “nulliparous,” as readers will expect this to include any delivery, regardless of mode. If the 41.3% is for any delivery, “vaginal” should be removed from the description.

4.       Table 3: Specifying “misoprostol dosage (after cesarean section)” is confusing. This covariate is present for all patients, correct? It’s just lower for those with a prior section? If so, it would be clearer to remove “(after cesarean section)” from the descriptor and leave the * with the note explaining this below.

Discussion

5.       Lines 179-181: This first specifies vaginal delivery without prior cesarean and then just vaginal delivery. Should this just be vaginal delivery, or was the combination of vaginal and cesarean deliveries taken into account?
